# A mouse-tracking study of the composite nature of the Stroop effect at the level of response execution

Boris Quétard[1,2]*, Nicolas Spatola[1,3], Benjamin A. Parris[4], Ludovic Ferrand[1], Maria Augustinova[5]*

**1** Université Clermont Auvergne, CNRS LAPSCO, Clermont-Ferrand, France, **2** Department of Brain and Cognition, University of Leuven (KU Leuven), Leuven, Belgium, **3** Artimon Perspectives, Paris, France, **4** Department of Psychology, Bournemouth University, Poole, United Kingdom, **5** Université de Rouen-Normandie, CRFDP, Rouen, France

* boris.quetard@gmail.com, boris.quetard@kuleuven.be (BQ); Maria.Augustinova@univ-rouen.fr (MA)

**Data Availability Statement:** The data presented in this study (including the raw data) as well as the R code of the statistical analyses presented in the article and in the Supporting Information files are

## Abstract

By forcing selection into response execution processes, the present mouse-tracking study investigated whether the ongoing process of response selection in the colour-word Stroop task is influenced by conflict and facilitation at both the level of response and stimulus. Mouse-tracking measures including partial errors provided credible evidence that both response and semantic conflict (i.e., distinct constituents of interference) contribute to the overall Stroop interference effect even after a response has been initiated. This contribution was also observed for the overall facilitation effect (that was credibly decomposed into response and semantic components in response times but not in mouse deviation measures). These results run counter to the dominant single-stage response competition models that currently fail to explain: 1) the expression of Stroop effects in measures of response execution and; 2) the composite nature of both interference and facilitation. By showing that Stroop effects–originating from multiple levels of processing–can cascade into movement parameters, the present study revealed the potential overlap between selection and execution process. It therefore calls for further theoretical efforts to account for when, where and under what conditions Stroop effects originating from different loci are controlled.

## Introduction

When performing the Stroop task [1], individuals are instructed to identify–as quickly and accurately as possible–the font colour of written stimuli while ignoring their actual meaning. Following these instructions is particularly challenging when the to-be-ignored word dimension of written stimuli denotes a different colour–as is the case in the emblematic colour-incongruent trials (e.g., the word "BLUE" displayed in yellow; hereafter $BLUE_{yellow}$). Indeed, colour-identification times are consistently longer for this type of trial compared to those observed for baseline trials (e.g., the word "DEAL" displayed in yellow, hereafter $DEAL_{yellow}$). This difference (e.g., $BLUE_{yellow}-DEAL_{yellow}$) is commonly referred to as a *Stroop effect*. The present study examined this effect in a mouse-tracking paradigm [2, 3].

available through the Open Science Framework at the webpage https://osf.io/f2cpk/. The webpage containing the data and code is now public. If this is necessary/preferable, we would like to register them properly (e.g., with a DOI). We are still figuring out how to do so.

**Funding:** The analysis and interpretation of data; the writing of the report; and the process related to submission of this article for publication was supported by Agence nationale de la recherche (ANR Grant ANR-19-CE28-0013) and Réseau d'Intérêt Normandie (RIN Tremplin Grant 19E00851) of Normandie Region, France awarded to senior authors (MA, LF & BP). The funders had no role in study design, data collection and analysis, decision to publish, or preparation of the manuscript. - BQ was paid on long-term structural funding from the Flemish Government to Pr. Johan Wagemans (METH/14/02 and METH/21/02). The funders had no role in study design, data collection and analysis, decision to publish, or preparation of the manuscript.

**Competing interests:** The authors declare that no competing interests exist.

## Response selection vs. execution processes in the colour-word Stroop task

Because word reading is routinized through practice, the irrelevant word dimension of Stroop stimuli often provides evidence towards an incorrect response (e.g., blue for $BLUE_{yellow}$). Since "blue" (unlike "deal") is included in the set of possible responses, the evidence in favour of the incorrect response, interferes with the one cued by the relevant colour dimension (i.e., yellow for $BLUE_{yellow}$). While this *response conflict* is assumed to be resolved at the level of response selection [4–10], only a handful of studies have addressed this assumption–shared by extant theories of the Stroop effect–directly.

To this end, Logan and Zbrodoff [11] instructed participants to indicate the font colour of Stroop stimuli by typing their response. In several experiments, they observed a large Stroop effect on response selection (i.e., time taken to press the first letter) but no Stroop effect on response execution (i.e., the time that elapsed between pressing the first and the last letter). This dissociation (i.e., Stroop effects on response selection but not on response execution) was found with other response modalities. Indeed, a Stroop effect was reported on vocal naming latencies but not on their durations [12] (Exp.1) and on saccadic latencies, but not on their amplitudes or velocity [13]. These results seem therefore consistent with the idea that response selection and response execution in the Stroop task are serial processes and that the aforementioned response conflict is successfully resolved before a response is executed.

However, in the abovementioned study Kello and colleagues [12] showed that when a response deadline was introduced, a Stroop effect on vocal naming latencies was substantially reduced (~70ms in Experiment 2 vs. ~110ms in Experiment 1) and a ~45ms Stroop effect on vocal naming durations emerged. They therefore argued that processes pertaining to selection and execution of the response can unfold in a cascaded (as opposed to serial) fashion (see [12] for definitions and discussions of these terms) depending on experimental context. Hand-tracking (i.e., mouse- and reach-tracking) has been shown to be useful in investigating this overlap between selection and execution processes [14–18], revealing significant Stroop effects on initiation times and on later response executive, revealing that response selection can continue to evolve even after a movement has been initiated such that it continues to deviate from the direct path towards a correct response (e.g., "yellow" for $BLUE_{yellow}$). This ongoing modification of responses might actually better reflect attentional selection in real life in which we might not have the luxury of acting only when a decision has been reached.

While these deviations occur independently of whether [18] or not [14–17] speeded-responding is introduced, it is important to understand that they do not necessarily reflect movements towards an incorrect response. Indeed, hand-movements simply directed towards the centre of the screen also generate movement deviations. Therefore, the extent to which hand-deviations measures reported in existing studies specifically reflect the co-activation of competing responses is unclear. To address this issue, the first goal of the present study was to supplement standard mouse-tracking measures with the analysis of response trajectories clearly directed (at any point of the movement) towards the incorrect response, but ending their course at the correct response (i.e., so-called *partial errors* that reflect the overlap between selection and execution processes unambiguously; see [19] for this type of analysis in a visually-guided reaching task).

## The composite nature of the Stroop effect

All the existing hand-tracking studies [14–18] used colour-congruent trials (e.g., the word "BLUE" displayed in blue; hereafter $BLUE_{blue}$) to derive their measure of interference on colour-incongruent trials ($BLUE_{yellow}$). However, the font colour of colour-congruent trials is known to be identified faster than that of colour-neutral trials (see [20], Exp. 2 for the first

demonstration). Therefore, considering the Stroop effect without colour-neutral trials (e.g., $BLUE_{yellow}-BLUE_{blue}$) fails to recognize the role that *facilitation* [21] plays in this composite effect. It therefore also fails to provide a reliable measure of interference (see [22, 23] for discussions). Indeed, as already emphasized by MacLeod in his seminal paper [24], the overall Stroop (or congruency) effect actually represents "(. . .) the sum of facilitation and interference, each in unknown amounts" (p.168). Consequently, the extent to which the ongoing process of response selection in the existing hand-tracking studies is influenced by interference (as opposed to facilitation) is at this point also unclear. To address this issue in the present study, interference and facilitation were considered separately in mouse-tracking measures.

A further motivation for this latter endeavour is that these distinct constituents of the overall Stroop (or congruency) effect can also be further decomposed. Some accounts of the Stroop effect anticipate that, in addition to the overlap occurring between stimuli and responses (i.e., overlap that generates response conflict, see previous section), colour (in)congruency also entails conceptual similarity between relevant and irrelevant stimulus sets of colour-incongruent words. This latter type of overlap generates a different type of conflict—so-called *stimulus* conflict [25–29].

In line with this general logic, it has been argued [25] that stimulus conflict is semantic in its nature (see also e.g., [30–35]). Under this view, (automatic) processing of the word-dimension of an incongruent trial interferes with processing of its colour-dimension precisely because the *meaning* of the former dimension overlaps with that of the latter (but see [36] for the idea of perceptual rather than conceptual/semantic interference at the stimulus level). Consequently, compared to a colour-neutral baseline ($DEAL_{yellow}$), the delay in processing (i.e., interference) occurs not only for colour-incongruent words depicted above (e.g., $BLUE_{yellow}$, hereafter *standard* colour-incongruent words) but also for so-called colour-associated words (e.g., *SKY*) presented in an incongruent colour ($SKY_{yellow}$) [35]. But only for standard colour-incongruent trials (e.g., $BLUE_{yellow}$), additional interference arises at the response level (i.e., response conflict; [25, 27]; but see [37]).

In line with this idea, Manwell and colleagues [38] reported a 149ms standard interference effect ($BLUE yellow-DEAL_{yellow}$) along with a significant 28ms semantic-associative interference effect (e.g., $SKY yellow-DEAL_{yellow}$). By applying this same logic to Stroop facilitation, Dalrymple-Alford [21] was the first to report a 67ms standard facilitation effect ($DEAL_{blue}-BLUE_{blue}$) along with a 42ms semantic-associative facilitation effect (e.g., $DEAL_{blue}-SKY_{blue}$). Since, numerous studies (see [23, 25, 39] for reviews) have suggested both the Stroop interference and facilitation effect can be themselves considered composite phenomena. Therefore, the third goal of the present study was to examine the extent to which the ongoing process of response selection in the colour-word Stroop task is influenced by conflict and facilitation at both the level of response and stimulus (see Fig 1).

## Present study

To address these distinct goals, participants were required to move a computer mouse from a starting position (located in the lower half of the screen) to one of the four colour-patches located in the upper half of the computer screen (see Fig 2). To maximise the direct measurement of ongoing response selection process in the Stroop task, an initiation deadline required participants to initiate their responses early [18]. In addition to initiation times, response times, maximum mouse deviations and evolvement of mouse deviation across normalized time (see [14] for a closely related measure, see also [40] for a methods article), partial errors [19] were collected in this setup. Unlike mouse deviation measures, partial errors–specifically directed towards the incorrect response but ending their course at the correct response–were

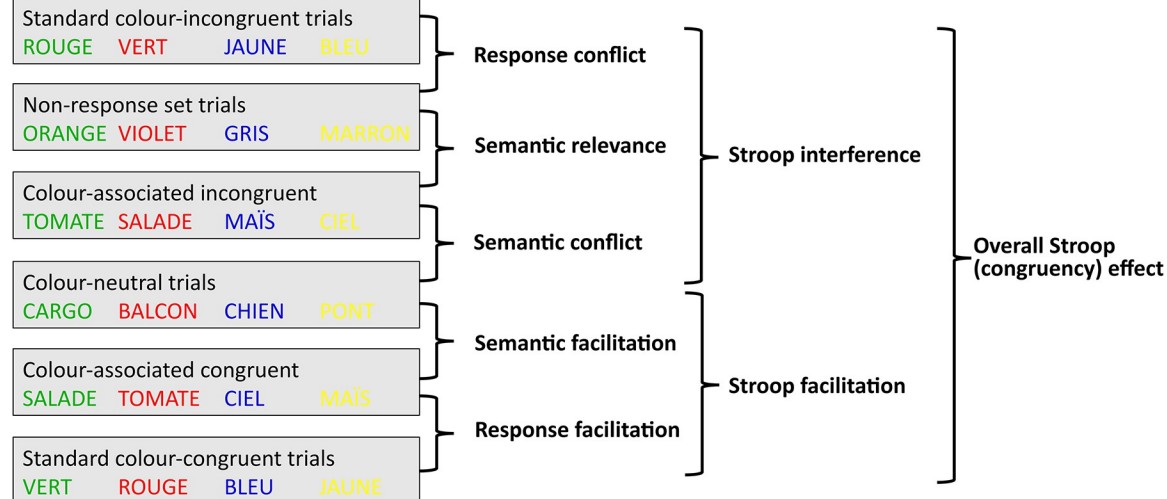

**Fig 1. Trial types employed and comparisons made in the experiment to enable the indexing of the different components of the overall (congruency) Stroop effect.** Ink colours are paired: Green with red, blue with yellow. For colour-neutral and non-response set conditions, the words can be written with each colours of their pair (e.g., "Brown" written either in yellow or blue). Stroop components are calculated by subtracting the lower condition from the upper condition.

expected to reflect the co-activation of two competing responses more directly than mouse deviations alone (first goal of the present study).

To investigate the respective contribution of Stroop interference [1] and facilitation [21] to the overall Stroop (or congruency) effect in mouse-tracking measures (the second goal of the present study), colour-neutral trials (e.g., $DEAL_{yellow}$) were included (see Fig 1). Specifically, the Stroop interference ($BLUE_{yellow}$–$DEAL_{yellow}$) was expected to be evidenced by a particularly strong *deviation* from the appropriate colour response for standard colour-incongruent trials

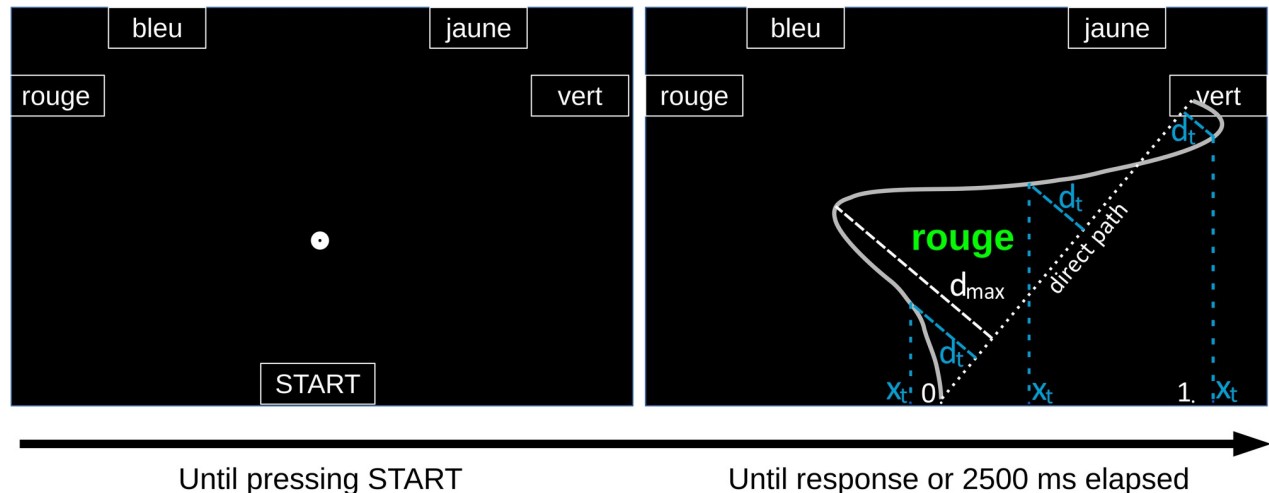

**Fig 2. Spatial layout of the experimental display and illustration of the mouse deviation and x-coordinates.** The example displays a standard incongruent item ("rouge" [red] displayed in green). The grey curve represents a possible mouse trajectory response; the oblique dotted white line represents the direct path from the start point to the response. $d_{max}$ denotes the (orthogonal) maximal deviation from the trajectory to the direct path with its corresponding orthogonal projection line on the direct path (long-dashed white line). Let t denote time sample numbers; $d_t$ represent the trajectory's (orthogonal) deviation at three time samples with their respective projection line (long-dashed blue lines); $x_t$ indicates the x-coordinate of the trajectory at those three time samples, with their respective projection lines on the x-axis (short-dashed blue lines). Due to the trajectories' alignment, the x-coordinate at the starting point of the trajectory is 0 (origin) and x-coordinate at end point (response) is 1 (displayed in the Figure). Negative x-coordinates are on the incorrect response side.

($BLUE_{yellow}$) as compared to their colour-neutral counterparts. Again, the extent to which this deviation is clearly directed to the incorrect response was expected to be seen in partial errors. Although the extent to which mouse deviation measures sensitively capture Stroop facilitation has not been previously investigated, the ongoing process of response selection for colour-congruent trial ($BLUE_{blue}$) could in principle result in a straighter trajectory (less deviated) compared to that of a colour-neutral trial ($DEAL_{blue}$).

Finally, the present study used mouse deviation measures to investigate the respective contribution of conflict and facilitation at both the level of response and stimulus to the overall Stroop effect (the third goal of the present study). To this end, participants were also asked to identify the font colour of colour-associated items (e.g., *SKY*) and that of so-called non-response set trials (e.g., *GREY*). In non-response set trials ($GREY_{yellow}$, see Fig 1), the irrelevant word dimension depicts a colour that is different from the one to be named and therefore generates stimulus or semantic conflict [41]. However, since this colour is not included in the response set (containing only blue, yellow, green and red), it is likely to be free of response conflict in the same way as colour-associated incongruent words ($SKY_{yellow}$, see previous section, see also [42–44]). This trial type–that is by definition incongruent–was therefore used as baseline against which the magnitude of response conflict ($BLUE_{yellow}$–$GREY_{yellow}$) was estimated, while the measure of semantic conflict was derived as a difference between colour-associated incongruent and colour-neutral trials ($SKY_{yellow}$−$DEAL_{yellow}$; [23, 37, 45]). While all conflict items ($BLUE_{yellow}$, $GREY_{yellow}$ and $SKY_{yellow}$) should produce *deviation* from the appropriate colour response (as compared to the relevant baseline), this deviation should be greater for items involving both semantic and response conflict (i.e., $BLUE_{yellow}$) as compared to those involving semantic conflict alone (i.e., $GREY_{yellow}$ and $SKY_{yellow}$). Even more importantly, if semantic conflict indeed reflects competing colour concepts (as opposed competing responses), these latter items should produce a lower rate of partial errors than those involving response conflict (i.e., standard colour-incongruent items).

It should be noted at this point that when response conflict and semantic conflicts are measured independently (e.g., by using different baselines), a part of the Stroop interference effect ($BLUE_{yellow}$–$DEAL_{yellow}$) often remains unexplained by the sum of the two conflicts. Therefore, in addition to semantic and response conflicts, the present study also examined the difference between non-response set and associated colour-incongruent words ($GREY_{yellow}$−$SKY_{yellow}$), often referred to as *semantic relevance* [43; see Fig 1].

Finally, given that non-response set trials ($GREY_{yellow}$) do not have colour-congruent counterparts, response facilitation was derived as the difference between colour-associated and standard colour-congruent items ($SKY_{blue}$−$BLUE_{blue}$) and semantic facilitation was derived as the difference between colour-neutral and associated colour-congruent items ($DEAL_{blue}$−$SKY_{blue}$). Again, the extent to which mouse-tracking measurements are actually sensitive to semantic (as opposed to response) conflict/facilitation is currently an unresolved issue. However, in line with the logic outlined above, a greater deviation from the appropriate colour response (as compared to the relevant baseline) was expected for conflict stimuli and stronger attractions towards the appropriate colour response (as compared to the relevant baseline) for facilitatory stimuli, however, partial errors–specifically reflecting deviations towards incorrect responses–were expected to occur mainly for response as opposed to semantic conflict.

## Method

### Participants and sample justification

Eighty-three (70 female and 13 male) undergraduates ($M_{age}$ = 20; $SD$ = 2.3), enrolled in the Psychology program at Université Clermont Auvergne (Clermont-Ferrand, France),

volunteered to participate in this study, which was approved by the institutional Research Ethics Committee of Université Clermont Auvergne (IRB000115-40-2019-18) and therefore carried out in accordance with Declaration of Helsinki (WMA, 2018). In line with the exclusion criteria, all were native French-speakers and reported normal (or corrected-to-normal) vision and colour-vision. Seventy-nine participants declared to be right-handed, three to be left-handed and one self-reported a bilateral handedness. However, only one left-handed and one right-handed participant preferred to use a computer mouse with their left hand. Given that there is, at least to our knowledge, no priori mouse-tracking study of the composite nature of the Stroop effect that can be used as guide for power simulations, the sample size was determined a priori following recommendations of Brysbaert and Stevens [46]. For a properly powered experiment with repeated chronometric measures, they suggest to collect at least 1600 observations per condition. Whilst a total sample size of 50 participants was necessary (given our number of trials per stimulus type), it was substantially increased to reliably estimate differences in all measures. Most particularly, partial errors frequencies cannot be estimated *a priori* due to the aforementioned lack of this type of published analyses in the Stroop task.

## Apparatus

OpenSesame [47] and the Mousetrap plug-in [48] were used to build and run a computerized version of the Stroop task. The stimuli were presented using a laptop computer connected to a 48×27 cm monitor (21.7-inch diagonal) at a resolution of 1920×1080 pixels. The participants responded using a regular computer mouse. Mouse movements were recorded at a 100-Hz frame rate and were pre-processed and analysed using the Mousetrap R package [48].

## Procedure

All the participants first read and signed an informed consent form and were asked to identify the colour of four patches (yellow, red, green, blue) printed on white paper to ensure that their colour vision was indeed normal. They were then seated in a darkened room approximately 70cm in front of the monitor and asked to place the mouse (on the right vs. left) in the position they preferred so that they could use the hand they were the most comfortable with to respond. After they had been familiarized with the general layout (see Fig 2), the participants were instructed to identify–as quickly and accurately as possible–the colour of the stimulus presented in the centre of the screen while ignoring everything else in the display. To this end, they were instructed to execute a mouse movement from the start box towards the appropriate response box. The participants were also informed about all the possible feedbacks they might see. More specifically, a "TIME OUT" message appeared if the participant's response took longer than the time limit (2500 ms) and an "X" sign appeared when it was false. When the response movement was initiated after 500 ms, the participants saw the message "Veuillez commencer votre mouvement de réponse plus rapidement s'il vous plait" (i.e., "Please start your response movement sooner"). The participants familiarized themselves with these requirements, namely the aforementioned response deadline, during 32 practice trials consisting of coloured rows of the letter "x". They then completed 192 experimental trials (see next section for detailed description) intermixed in a single experimental block. After completing the experimental block, the participants were thanked and left the laboratory.

## Layout, stimuli and design

As in Bundt and colleagues' study [18], the general display consisted of four response boxes labelled (in bold lowercase Arial font, size 51 pixels, i.e., 1.275cm) "rouge", "jaune", "bleu", "vert", *i.e.*, the French words for "red", "yellow", "blue", and "green"), and one start

box (labelled "START" in uppercase Arial font, size 40 pixels, i.e., 1cm). The boxes were delimited by white contours on a black background and there was also a fixation dot indicating the future location of the stimulus (see Fig 2). Participants initiated stimulus presentation by clicking with the mouse cursor on the start box (Fig 2). When they did this, the cursor position automatically shifted to the centre of the start box and one colour-stimulus appeared approximately 3.18cm below the central point of the screen (but centred on the x-axis) for a maximum duration of 2500ms (Fig 2). Said differently, clicking centred the cursor so each trial started at the same exact point.

The stimuli, which consisted of four (French) colour-words–*rouge* [red], *jaune* [yellow], *bleu* [blue], and *vert* [green] and four colour-associated words: *tomate* [tomato], *maïs* [corn], *ciel* [sky], and *salade* [salad],–were presented in congruent and incongruent colours (in bold lowercase Arial font, size 100 pixels, i.e., 2.5cm). Four additional (French) colour-words–violet [purple], marron [brown], gris [grey], and orange [orange]–were used to create non-response set stimuli and therefore only appeared in (response-set) colours (i.e., colours that were incongruent with their actual meaning). Finally, four colour-neutral words–balcon [balcony], cargo [cargo], pont [bridge] and chien [dog]–also appeared in (response-set) colours. It should be noted that the *red* and *green* and the *blue* and *yellow* response boxes, respectively, were paired in such a way that they appeared at mirroring positions around the upper corners of the computer screen. However, their respective positions on the x- and y-axis were counterbalanced across participants and thus appeared in four possible configurations. To control further for the geometrical and probabilistic symmetry between correct and incorrect responses, the standard and colour-associated words only appeared in either a congruent or in the incongruent colour they were paired with. Thus, for instance, green/salad were always presented in green (congruent) or red (incongruent) and blue/sky were always presented in blue (congruent) or yellow (incongruent). Colour-neutral and non-response set words were paired accordingly (the order in which they are presented in the preceding paragraph reflects this). Therefore, purple/balcony (paired with red and green) for instance could be presented in red or green, whereas brown/bridge (paired with yellow and blue) could be displayed in blue or yellow. The repetition of these pairings resulted in 32 trials in each of the 6 conditions of a single (i.e., Stimulus-type) factor that governed data collection (i.e., 192 experimental stimuli in total intermixed in a single experimental block).

## Results

### Data processing and analysis

Trials with no (0.34% of the total data) or incorrect (0.58% of the total data) responses were excluded from further analyses (see S3 File for a table displaying the mean error and omission rates per condition) and so were response times more than 3*SD*s above or below each participant's mean latency for each condition (i.e., 1% of the total data). This filtering procedure has the advantage of taking out extreme values without affecting the data of one condition or of one participant in particular. Given that there were two symmetrical (and matched) response boxes on each side of the screen (see Method), mouse trajectories were remapped rightward and spatially aligned prior to computing the different mouse-tracking measures. This ensured that despite spatial variations due to differences in the position of response boxes, the trajectories had the same spatial coordinates at their start- and end-points (start: x = 0, y = 0; end: x = 1, y = 1.5) and were thus comparable. Note that upon clicking the Start button, the mouse cursor was also automatically centred. Initiation Times was defined as the time that elapsed between clicking the start box and the beginning of the mouse movement; Response Times was defined as the time that elapsed between clicking the *Start* box and selecting an appropriate response box; Maximum Deviations

was defined as the maximum orthogonal distance from any point on the trajectory of the mouse movement to the direct path (i.e., the straight line from the start- to end-point of the trajectory), Fig 2 displays the direct path (dotted white line) and Maximum Deviation (long-dashed white line). Those measures were computed using the Mousetrap R package (v3.1.4; [48]). It should be remembered that Maximum Deviation can be calculated above the direct path (positive deviation), below (negative deviation), or both (absolute deviation). In Fig 2, one example of negative deviation is given (deviation calculated at the third latest time sample $d_t$, displayed in blue). If at one point, the mouse trajectory strongly deviates towards the incorrect response–indicating coactive responses–it is more sensitively reflected in Maximum Deviations above the direct path that were therefore used here. The changes in the orthogonal mouse deviation and the x-coordinates of mouse trajectories across normalized time samples were also examined. As previously indicated, for time-based orthogonal mouse deviation analyses, a positive sign denotes a deviation above the direct path and a negative sign a deviation below the direct path (in Fig 2: positive deviations $d_t$ at first and second time samples and negative deviation $d_t$ at third time sample). While the orthogonal mouse deviation from the direct path takes into account the deviation on both x and y axes, the y-axis is less informative regarding the competition. Thus, the x-coordinates of the mouse trajectory points index more specifically the deviation towards a response side at each normalized time sample. A negative x-coordinate indicates a movement that is deviated towards the incorrect response (given that trajectories are remapped rightward, and the start point/centre of the x-axis is 0, see Fig 2). The mouse trajectories were time-normalized into 101 time-steps (from 1 to 101) with the result that testing the difference in mouse deviation and x-coordinate between two conditions (e.g., colour-congruent vs. colour-incongruent items) at time step 11 meant that this difference was estimated at the time point corresponding to approximately 10% of the total duration of each response movement (including the time that elapsed before movement initiation).

To identify trials with response trajectories directed towards the incorrect response but ending their course at the correct response (i.e., partial errors) and estimate their proportion for each type of stimulus, a data driven approach using clustering analysis was used (see S4 File for a detailed description). The mouse trajectory clustering analysis tool implemented in Moustrap R package [49] was used to this end. Namely, the mt_cluster() function was applied to group different input trajectories into eight clusters, depicted in Fig 3. As advised by the authors of this package, we used the hierarchical cluster algorithm implemented in the package on space-normalized trajectories (into 100 data points evenly spaced). This allowed the clustering of trajectories and the identification of profiles where the movements reached a point close to an incorrect response but were corrected late over the course of a trial (Cl4), those with movements directed to the incorrect response side (Cl8) although to lesser extent than in Cl4, as well as those with a movement initially directed towards the correct response, then wrongly corrected towards the incorrect responses before being finally re-corrected (Cl3). The overall rate of these partial errors (clusters Cl3, Cl4 and Cl8, see Fig 3)–estimated to be ~27.1% of all considered mouse trajectories–was therefore distinguished from trajectories simply directed to the centre of a display (see Cl5 for instance; see also S4 File full details of partial error rates analyses conducted with the present; analyses with other numbers of clusters are also available on the OSF webpage). Proportions of partial errors (henceforth, PE) were then estimated for each condition of the Stimulus-type factor and for each participant (see Table 1). The differences in PE rates between conditions were analysed using linear mixed models.

## Linear mixed modelling

As in Quétard and colleagues' study [50], a Linear Mixed Modelling (LMMs) approach was used for data analysis. The lme4 R package (v1.1.23, [51]) was used to fit the models and the

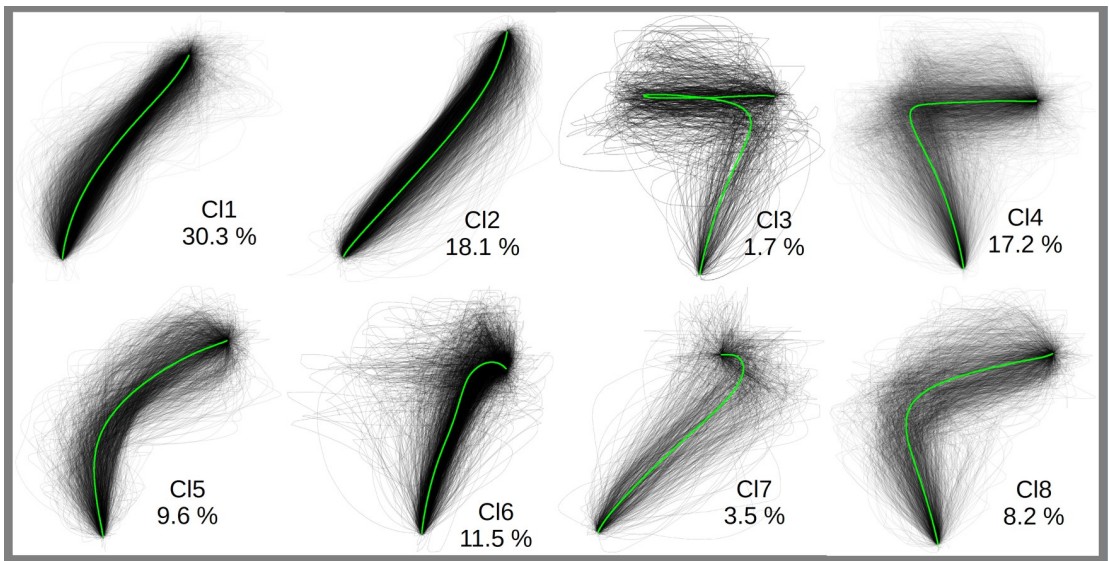

**Fig 3. Distribution of mouse trajectories (observed for all conditions of the Stimulus-type factor) across the clusters (Cl) estimated through hierarchical cluster analysis.** The green lines represent each cluster's average trajectory. Cl3, Cl4 and Cl8 are considered as partial error clusters for the purpose of estimating partial error rates.

emmeans R package (v1.4.6) was then used to estimate the marginal means. To this end, the six conditions of Stimulus-type variable were contrasted into dummy variables with colour-neutral items as reference level and the five resulting parameters as fixed effects. A general method derived from Bates et al. [52] and Matuschek et al. [53] was used to select a parsimonious random effect structure. This procedure was applied manually to model measures of Initiation and Response Times along with that of Maximum Deviation. As each partial error rate was calculated per condition and participant, this model selection procedure was impossible since only the random intercept could be estimated.

**Table 1. Marginal means and standard errors (SE) estimated using linear mixed modelling for initiation times, response times, maximum deviation and partial errors as a function of Stimulus type.**

| | Initiation time (ms) | | Response time (ms) | | Maximum Deviation | | Partial error rates (%) | |
|---|---|---|---|---|---|---|---|---|
| Stimulus-type | Mean | *(SE)* | Mean | *(SE)* | Mean | *(SE)* | Mean | *(SE)* |
| **Standard colour-incongruent** $BLUE_{yellow}$ | 169 | (7.5) | 1189 | (18.9) | 0.676 | (0.028) | 36.7 | (1.39) |
| **Non-response set** $GREY_{yellow}$ | 169 | (7.5) | 1111 | (14.9) | 0.517 | (0.02) | 26.1 | (1.39) |
| **Colour-associated incongruent** $SKY_{yellow}$ | 168 | (7.5) | 1075 | (13.2) | 0.526 | (0.021) | 28.6 | (1.39) |
| **Colour-neutral** $BRIDGE_{yellow}$ | 168 | (7.5) | 1070 | (13.2) | 0.480 | (0.019) | 24.9 | (1.39) |
| **Colour-associated congruent** $SKY_{blue}$ | 170 | (7.5) | 1045 | (13.2) | 0.459 | (0.019) | 24 | (1.39) |
| **Standard colour-congruent** $BLUE_{blue}$ | 167 | (7.5) | 1029 | (13.2) | 0.427 | (0.019) | 22.3 | (1.39) |

Note: Equivalent SEs across conditions are due to random slopes contrasting colour-neutral and some other stimulus types being excluded from the linear mixed model. Then, the SE estimated for those stimulus types correspond to the model intercept's SE (colour-neutral condition).

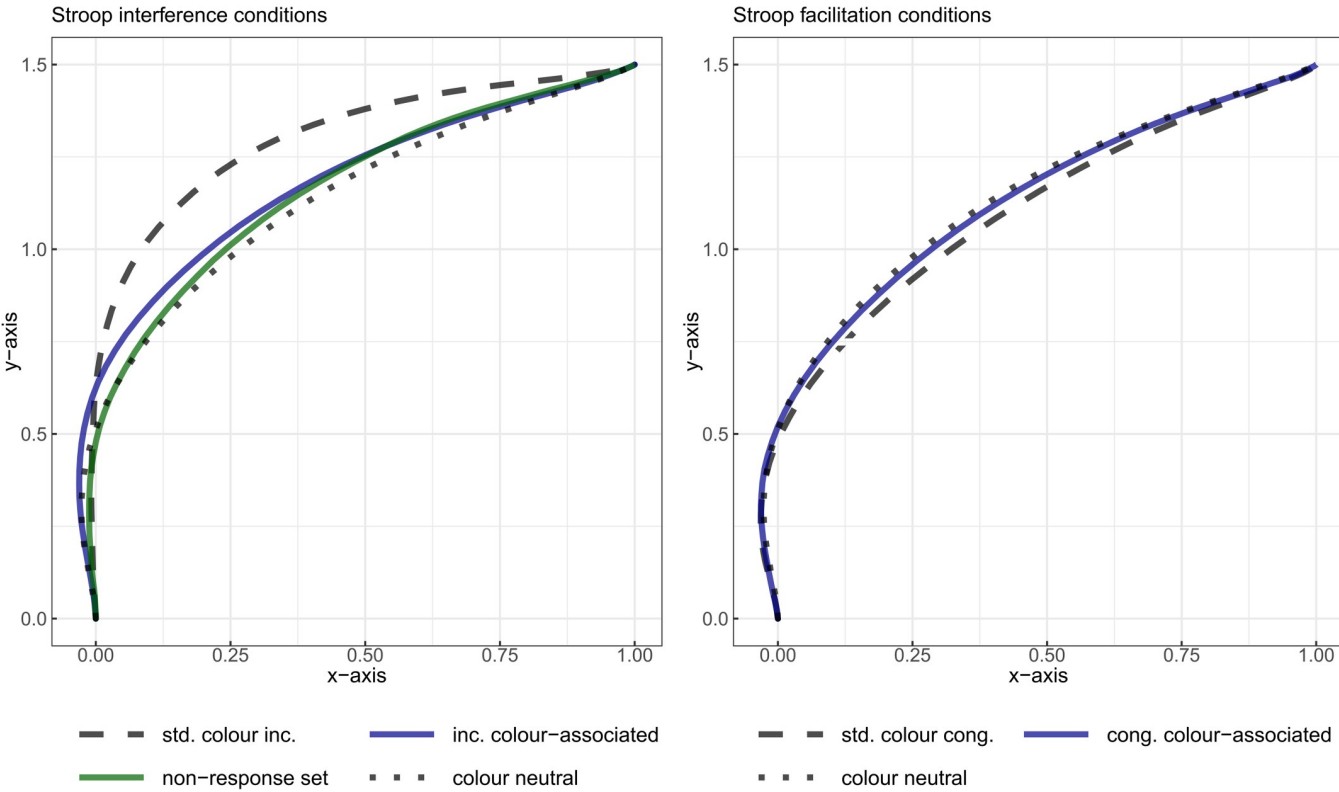

**Fig 4. Averaged time-normalized mouse trajectories aggregated by Stimulus-type.** Panel A: Stimulus-type conditions used for estimating interference components. Panel B: Stimulus-type conditions used for estimating facilitation components. To ensure comparability across trajectories, they were remapped rightward and their start/end points were aligned before time-normalization and aggregation by Stimulus-type and participant. Abbreviations: std.: standard, inc.: incongruent, cong: congruent.

To model the spatial trajectory measures across normalized time (i.e., deviation and x-coordinate), a simplified version of this model selection method was applied automatically at each time step using a custom algorithm (see S1 File for additional details of the procedure; 2 and 4 for a report of the estimated models' parameters for different measures). When the final model was fitted, the marginal means for each Stimulus-type were estimated (see Table 1 for descriptive statistics provided for the global trajectory measures, and Fig 4 for the time-normalized mouse trajectories aggregated by experimental conditions related to interference (Fig 4A) and facilitation (Fig 4B). Finally, the differences between the marginal means of the Stimulus-type conditions–relevant for decomposing the total Stroop effect into its different components– were estimated. In line with Amrhein and colleagues [54], this statistical modelling approach focused on parameter estimation instead of null-hypothesis significance testing. However, 99% confidence intervals (calculated with the Satterthwaite degrees of freedom approximation and alpha = 0.01) that do not include zero can be interpreted as "statistically significant". Using 99% confidence intervals allows for correcting for multiple comparisons from 95% confidence intervals; Indeed, using alpha = 0.01 is the Bonferroni correction for 5 comparisons (components) with alpha = 0.05, which is the number of lower-rank comparisons we make (the 3 other components, i.e., Stroop effect, Interference and Facilitation encompass the other components, see Fig 1). In our case, using 99% CI led to more conservative decisions than using Holm correction. For easier comparisons with previous studies on the Stroop effect, the same data analysed with traditional repeated measures ANOVA and paired comparisons are

provided in S5 File (along with effect sizes estimates and an a priori power/sensitivity analysis relevant for such analyses). We also provide analyses of the between- and within-participant variability of the Stroop components in S6 File and discuss their limitations (the present experiment is not built for analysing those efficiently)

Finally, the analysis of the Stroop effect and of its components across normalized time focused on time intervals where zero was outside the 95% confidence interval for more than 10 consecutive time steps (10% of total movement duration) in order to compensate for multiple comparison testing [55], and therefore, we do not correct the alpha threshold (i.e., we use alpha = 0.05–95% CI).

## The total Stroop effect (The Stroop congruency effect)

The Initiation Times (see Table 1) for standard colour-incongruent items were estimated to increase by 2.5ms (SE = 3.09) compared with those for standard colour-congruent items, with credible values of this difference ranging from -5.4ms to 10.5ms (*99% CI, df* = 15531). Given this absence of results (see Table 1 for their marginal means estimated through the model), Initiation Times were not analysed further (but see S2 File for further information). In Response Times (RTs), the total Stroop effect reached a mean magnitude of 159.8ms (*SE* = 12.5) with a 99% confidence interval ranging from 127.1ms to 192.6ms (*df* = 100, see Fig 5A). Compared to standard colour-congruent items, standard colour-incongruent items also induced a maximum geometrical deviation (MD) of 0.249 (*SE* = 0.026), with a large range of credible values (*99% CI* = [0.182, 0.316], *df* = 104.1; Fig 5B).

This difference in deviations of mouse-trajectories evolved across normalized time and peaked at time step 62 (i.e., 61% of the total response duration) with a magnitude of 0.245 (*SE* = 0.024, *95% CI* = [0.197, 0.293], *df* = 101.6, see Fig 6A, 6B). If the lower bound of the confidence intervals is taken as the cut-off value, this means that the total Stroop effect in mouse deviations considered across normalized time started to diverge from zero at time step 12. The total Stroop effect in the x-coordinate reached its peak at time step 64 with a magnitude of -0.267 (*SE* = 0.027, *95% CI* = [-0.319, -0.214], *df* = 100.7, see Fig 6C, 6D). Given that trajectories

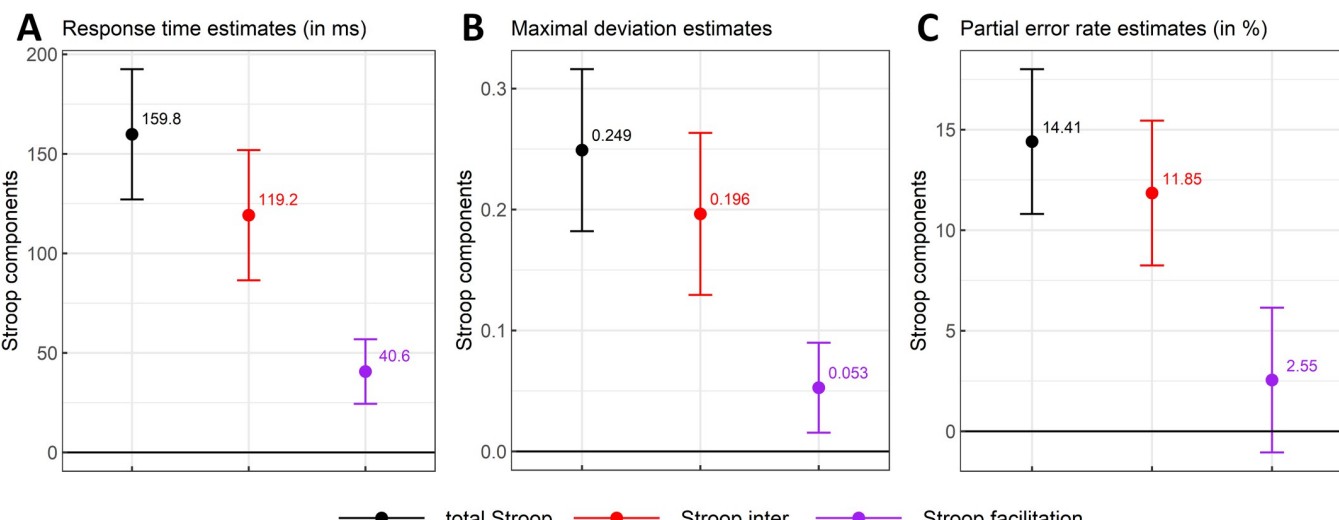

**Fig 5. Total Stroop effect (standard colour-incongruent–standard colour-congruent means) with Stroop interference (standard colour-incongruent– colour-neutral means) and facilitation (colour-neutral–standard colour-congruent means) components as a function of response times (Panel 5A), maximal mouse deviation (Panel 5B) and partial error rates (Panel 5C).** The magnitude and 99% CI of each component were estimated by contrasting the marginal means (i.e., marginal effects) of two types of stimuli computed from a fitted linear mixed model.

were remapped rightward (see Figs 2–4) with the centre of the x-axis set to zero, values for Stroop effects estimated with x-coordinates are generally negative. This means that on the x-axis, mouse trajectories for standard colour-incongruent items deviated by 0.267 units more towards the concurrent (i.e., incorrect) response side, compared to mouse trajectories for standard colour-congruent items. Confidence intervals for these differences in deviations excluded zero from time step 46 to 93. Beforehand, a very slight deviation on the x-axis towards the correct response was observed between time steps 30 and 40, reaching a magnitude of 0.029 at time step 37 (*SE* = 0.009, *95% CI* = [0.011, 0.046], *df* = 195.3). In line with the idea that these latter mouse deviations effects reflect–at least partly–the co-activation of competing responses, partial errors (PE) increased by 14.41% (*SE* = 1.39, *99% CI* = [10.81, 18.01], *df* = 410) for standard colour-incongruent words compared to standard colour-congruent words (Fig 5C).

In sum, as in past studies, the total (congruency) Stroop effect was found in different mouse measures [14–17] except Initiation Times [18]. Notably however, the present study provides the first evidence for a total colour-word Stroop effect in partial errors.

## Decomposing the total Stroop effect into interference vs. facilitation effects

The extent to which the aforementioned total Stroop effect (of sizeable magnitude in all measures except ITs) reliably results from the contribution of both interference (standard colour-incongruent–colour-neutral items) and facilitation (colour-neutral–standard colour-congruent items; see Fig 4A, 4B for their respective mouse trajectories) was further examined.

**The Stroop interference effect.**   RTs for standard colour-incongruent items were estimated to increase by 119.2 ms (*SE* = 12.5) compared to those for colour-neutral items, with credible values ranging from 86.5ms to 151.9ms (99% CI, *df* = 99.9; see Fig 5A). Standard colour-incongruent items also induced a maximum mouse deviation increase of 0.196 (SE = 0.026) compared to their colour-neutral counterparts. This latter difference in mouse deviations represented the most substantial part of the total Stroop effect (see Fig 5B). Even thought the 99% confidence interval for the Stroop interference effect in MDs was rather large, ranging from 0.129 to 0.263 (*df* = 104), its lower bound represented a substantial difference in terms of mouse deviation. This difference also evolved across time such that it reached its peak at time step 70 (i.e., 69% of the total response duration), with a magnitude of 0.202 (*SE* = 0.024, *95% CI* = [0.154, 0.25], *df* = 100.4, see Fig 6A). Again, if the lower bound of the confidence intervals is taken as the cut-off value, this means that the Stroop interference effect in mouse deviations considered across normalized time separated from zero between time step 14 and 92. Finally, the Stroop interference effect in the x-coordinate reached its peak at time step 64 with a magnitude of -0.217 (*SE* = 0.027, *95% CI* = [-0.269, -0.164], *df* = 100.6, see Fig 6C), with confidence intervals excluding zero from time step 47 to 92. Additionally, the trajectories deviated towards the correct response side on the x-axis early on, from time steps 32 to 38, but the number of subsequent time steps where 0 was excluded from the confidence interval did not reach 10. Crucially again, these latter effects in mouse deviation measures coincide with an increase of the PE rate of 11.85% (*SE* = 1.39, *99% CI* = [8.25, 15.45], *df* = 410, see Fig 5C) for standard colour-incongruent words compared with colour-neutral words.

**The Stroop facilitation effect.**   RTs for standard colour-congruent items were estimated to decrease by 40.6ms (*SE* = 6.3) compared to those for colour-neutral items, with credible values varying between 24.4ms and 56.9ms (*99% CI*, *df* = 15366.5; see Fig 5A). Compared to standard colour-congruent items, colour-neutral items induced a maximum increase in mouse deviation of 0.053 (*SE* = 0.014, see Fig 5B), with a *99% CI* ranging from 0.016 to 0.09 (*df* = 15288.2) and this facilitation effect represented 21.2% of the total Stroop effect. Again, it evolved across time such that it peaked at time step 58, with a magnitude of 0.048 (*SE* = 0.014,

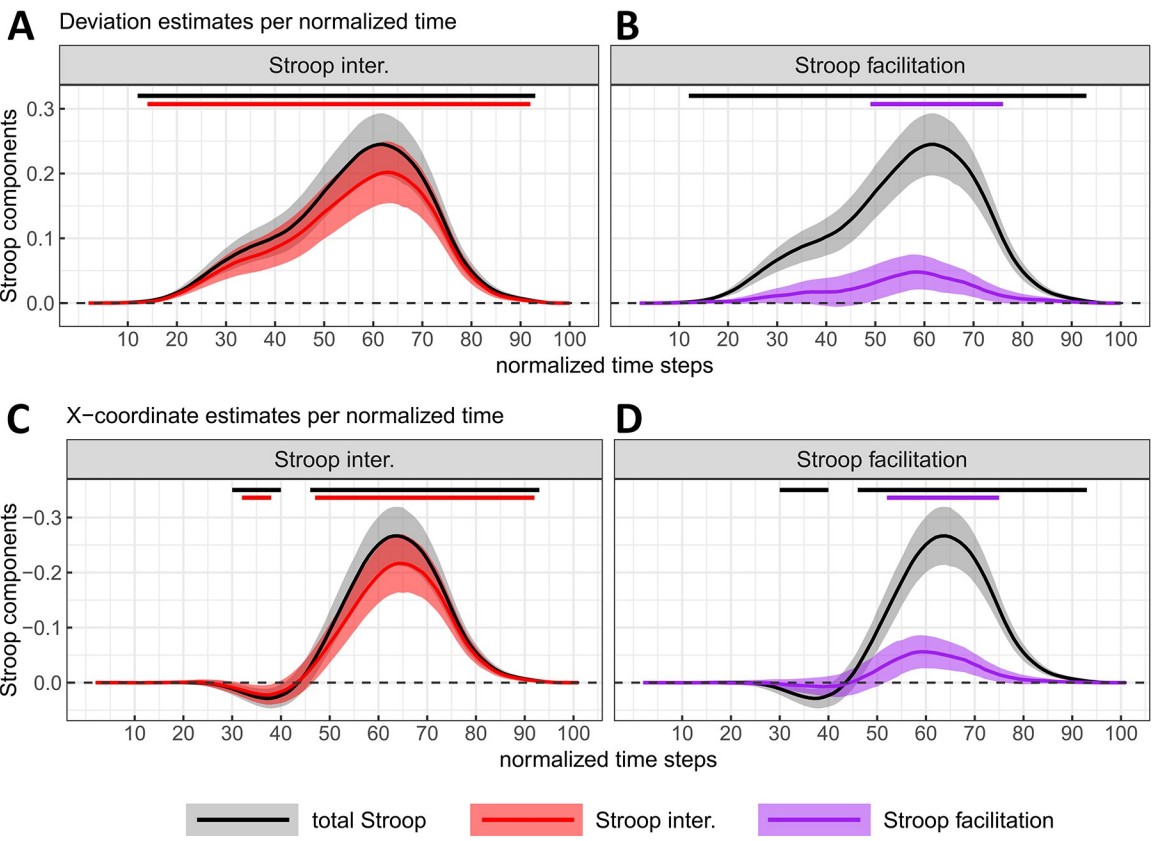

**Fig 6. Total Stroop effect with interference (Panels A-C) and facilitation (Panels B-D) components as a function of normalized time.** These are estimated from the mouse deviation towards the incorrect side (A, B) and the X mouse coordinate (C, D). The Total Stroop effect is displayed in all plots for visual comparison purposes. Negative X-coordinates indicate a deviation towards the incorrect response (the plot's y-axis is inverted for consistency with the other plot). The magnitude and 95% CI (shaded areas) of each component were estimated by contrasting the marginal means of two types of stimuli computed from a fitted linear mixed model (one LMM /time step and /measure, see the main text or Fig 5 caption for details of the contrasts). The straight lines at the top indicate the time steps where zero is not included in the 95% CI for at least 5 time-steps (in the main text, only intervals of at least 10 time-steps are reported).

*95% CI* = [0.021, 0.075], *df* = 15367, see Fig 6B). Confidence intervals excluded zero between time steps 49 and 76. On the x-coordinate, the deviation towards the incorrect response side peaked at time step 69 with a magnitude of -0.056 (*SE* = 0.015, *95% CI* = [-0.086, -0.026], *df* = 15366.9, see Fig 6D). Confidence intervals excluded zero from time step 52 to 75 (i.e., 24% of movement duration). In line with the idea that colour-neutral trials do not activate concurrent responses, these changes in mouse deviations are unlikely due to PE rates. Indeed, this facilitative effect could not be distinguished from zero, with a mean difference of 2.55% (*SE* = 1.39) and credible values ranging from -1.05% to 6.15% (*99% CI, df* = 410; see Fig 5C).

These results are therefore clearcut when it comes to relative contribution of both interference and facilitation to the total Stroop effect depicted above.

## Decomposing the Stroop interference effect into different conflicts

The same mouse-tracking measures were further used to address the contributions of response conflict (standard colour-incongruent–non-response set items), semantic conflict (colour-associated incongruent–colour-neutral items) and of semantic relevance (non-response set–colour-associated incongruent items) to the overall Stroop interference effect.

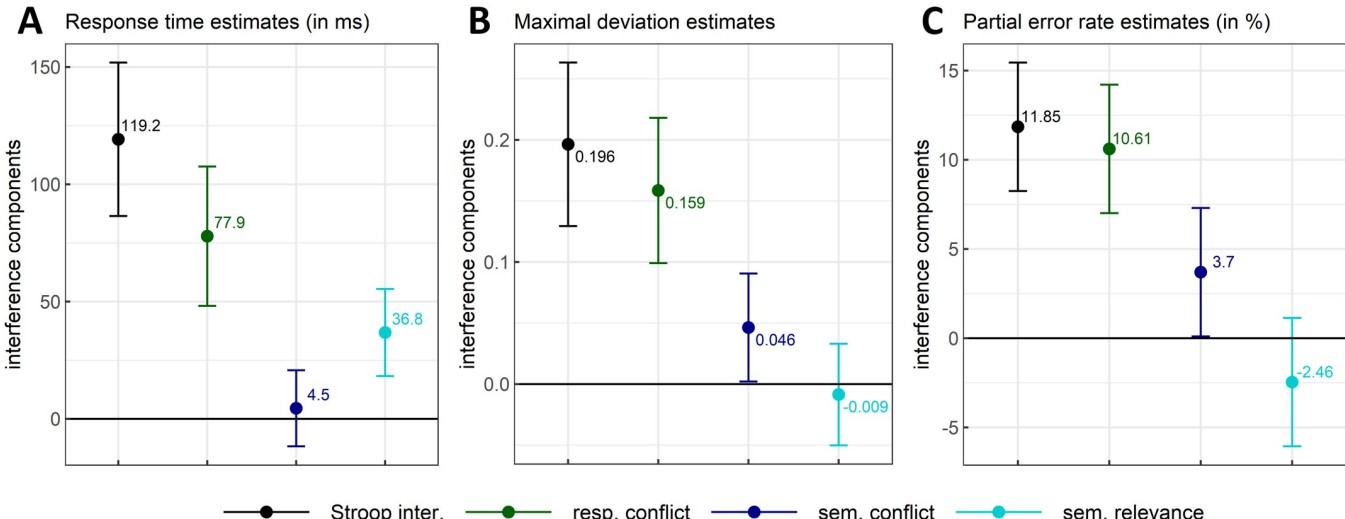

**Fig 7. Stroop interference (standard colour-incongruent–colour-neutral means) with response conflict (standard colour-incongruent–non-response set means), semantic conflict (colour-associated incongruent–colour-neutral means) and semantic relevance (non-response set–colour-associated incongruent means) components as a function of response times (Panel A), maximal mouse deviation (Panel B) and partial error rates (Panel C).** The magnitude and 99% CI of each component were estimated by contrasting the marginal means (i.e., marginal effects) of two types of stimuli computed from a fitted linear mixed model.

**Response conflict.** RTs for standard colour-incongruent items were estimated to increase by 77.9 ms ($SE$ = 11.3) compared with those for non-response set items (i.e., items that are free of response conflict), with credible values ranging from 48.2ms to 107.6ms (*99% CI, df* = 81.5; see Fig 7A). Compared to non-response set items, standard colour-incongruent items also induced a deviation of 0.159 ($SE$ = 0.023), with credible values ranging from 0.099 to 0.218 (*99% CI, df* = 82.1; see Fig 7B). This difference evolved across time such that it reached its peak at time step 61 with a magnitude of 0.162 ($SE$ = 0.024, *95% CI* = [0.114, 0.21], *df* = 103; see Fig 8A) and a confidence interval excluding 0 over most of the movements' duration (time steps 14 to 91). Response conflict in the x-coordinate reached its peak at time step 62 with a magnitude of -0.168 ($SE$ = 0.026, *95% CI* = [-0.221, -0.116], *df* = 103.8, see Fig 9A), with the confidence interval excluding zero from time steps 45 to 91. Finally, these latter effects on mouse deviations occurred in concert with changes on the PE rate–changes that represented almost the totality of Stroop interference with a mean of 10.61% ($SE$ = 1.39) and a *99% CI* ranging between 7.01 and 14.21% (*df* = 410; see Fig 7C). This substantial increase is in line with the idea that these items (unlike their non-response set counter-parts) generate response conflict. This idea that is reinforced further by the fact that non-response items caused approximately as many PEs as the colour-neutral items (see Table 1).

**Semantic conflict.** In the case of RTs, semantic conflict could not be distinguished from zero. Indeed, compared to colour-neutral items, colour-associated incongruent items were estimated to increase RTs by only 4.5ms ($SE$ = 6.3), with credible values ranging from -11.7 ms to 20.7 ms (*99% CI, df* = 15366.6; see Fig 7A). Nevertheless, compared to colour-neutral items, colour-associated incongruent items induced a slight deviation of 0.046 ($SE$ = 0.017) with credible values ranging from 0.002 to 0.091 (*99% CI, df* = 147.3; i.e., the lower value fell close to 0 but without including it; see Fig 7B). As a result, in this latter measure, semantic conflict contributed credibly to 6.5% to 40.7% (central value of 23.6%) of the overall Stroop interference effect.

The aforementioned difference in mouse deviations produced by colour-associated incongruent items compared to colour-neutral items evolved across time such that it reached its

Deviation estimates per normalized time

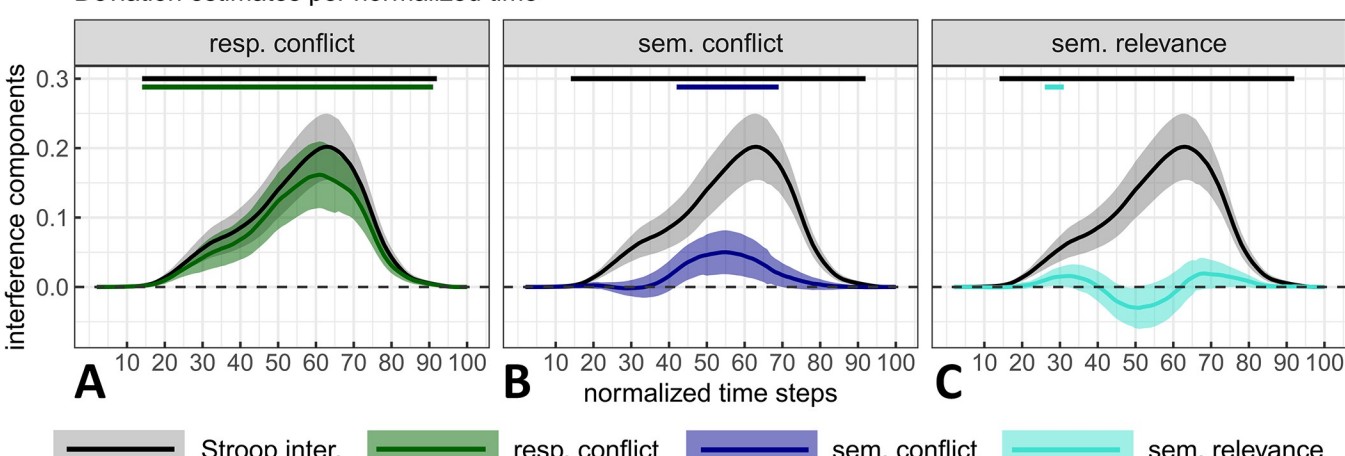

**Fig 8. Stroop interference with response conflict (Panel A), semantic conflict (Panel B) and semantic relevance (Panel C) components estimated from the mouse deviation towards the incorrect side as a function of normalized time.** Stroop interference is displayed in all plots for visual comparison purposes. The magnitude and 95% CI (shaded areas) of each component were estimated by contrasting the marginal means of two types of stimuli computed from a fitted linear mixed model (one LMM /time step and /measure, see main text or Fig 7 caption for a detail of the contrasts). Top straight lines indicate the time steps where zero is not included in the 95% CI for at least 5 time-steps (in the main text, only intervals of at least 10-time steps are reported).

peak at time step 55 with a magnitude of 0.05 ($SE$ = 0.016, $95\% CI$ = [0.018, 0.082], $df$ = 164.1; see Fig 8B). The confidence intervals excluded zero over a much smaller range (i.e., from time step 42 to 69), which represents 28% of total movement duration. In the x-coordinate, semantic conflict reached its peak at time step 55 with a magnitude of -0.053 ($SE$ = 0.017, $95\% CI$ = [-0.086, -0.019], $df$ = 175.5, see Fig 9B) and with the confidence intervals excluding zero from time step 43 to 70. Although in relatively modest extent, these changes in mouse deviations are also due to PE. Indeed, there increased by 3.7% ($SE$ = 1.39, $99\% CI$ = [0.01, 7.3], $df$ = 410) for colour-associated incongruent items as compared to colour-neutral items (see Fig 7C).

**Semantic relevance.** Despite this effect of 36.8ms ($SE$ = 7.1) on RTs (with credible values ranging from 18.2ms to 55.4 ms; $99\% CI$, $df$ = 163.9; see Fig 7A), non-response set incongruent items only induced a deviation of -0.009 ($SE$ = 0.016) compared to colour-associated incongruent items, with credible values ranging from -0.05 to 0.033 ($99\% CI$, $df$ = 81.5). Therefore, unsurprisingly, we could not identify a time interval (of min. 10-time steps) where estimates could be distinguished from zero in mouse deviations observed across normalized time (see Fig 8C). In the x-coordinate, semantic relevance reached a dip (maximal positive value) at time step 50 with a magnitude of 0.057 ($SE$ = 0.014, $95\% CI$ = [0.029, 0.085], $df$ = 15390.9, see Fig 9C) and with the confidence intervals excluding zero from time step 35 to 57. Positive semantic relevance values within this time interval indicate that the trajectories for colour-associated incongruent items were more directed towards the incorrect response side than those observed for non-response set items (since semantic relevance was calculated as non-response set minus colour-associated incongruent). This latter observation is reinforced by PE-rates. It decreased by -2.46% on average ($SE$ = 1.39), between colour-associated incongruent items and non-response items but the $99\% CI$ included 0 ($99\% CI$ = [-6.05, 1.14], $df$ = 410; Fig 7C).

## Decomposing the Stroop facilitation effect

Mirroring the previous analyses, the extent to which the Stroop facilitation effect results itself from the specific contribution of semantic facilitation (colour-neutral–colour-associated

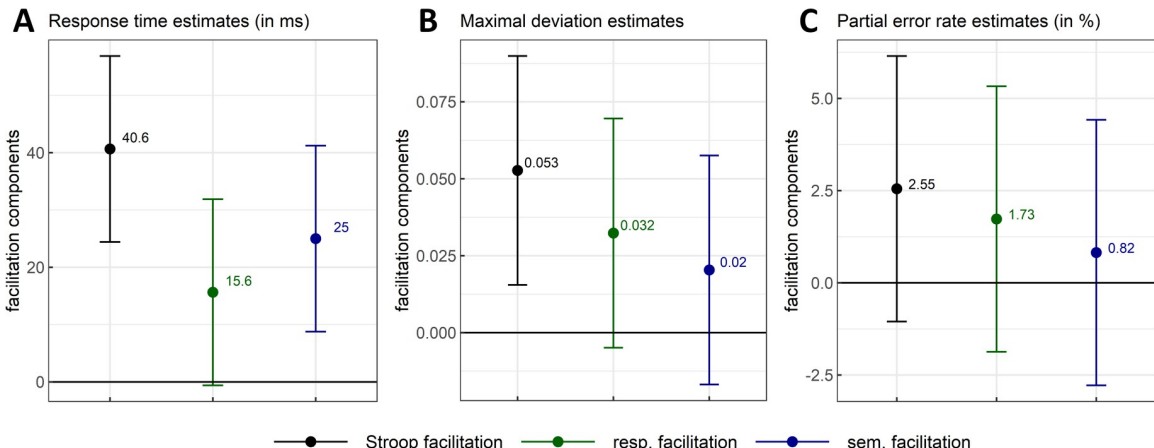

**Fig 9. Stroop interference with (Panel A), semantic conflict (Panel B) and semantic relevance (Panel C) estimated from the X mouse coordinate as a function of normalized time.** Negative X-coordinates indicate a deviation towards the incorrect response (plot's y-axis is inverted for consistency with the other plot). See Fig 8 caption for additional details.

congruent items) and of response facilitation (colour-associated congruent–standard colour-congruent items) was further examined.

**Semantic facilitation.** The results show a 25ms (*SE* = 6.3) facilitative effect of colour-associated congruent items (as compared to their colour-neutral counter-parts) in RTs with credible values ranging from 8.7ms to 41.2ms (*99% CI, df* = 15366.5; see Fig 10A). However, the estimations on remaining measures were too uncertain to confidently exclude the possibility that the corresponding values of these effects were null or negative (see Fig 10B, 10C for MD and PE respectively and Fig 11B and 11D for evolutions of mouse deviation and X mouse coordinates across normalized time).

**Response facilitation.** RTs for standard colour-congruent items were estimated to decrease by 15.6ms (*SE* = 6.3) compared to those for colour-associated congruent items, but

**Fig 10. Stroop facilitation (colour-neutral–standard colour-congruent means) with response (colour-associated congruent–standard colour-congruent means) and semantic (colour-neutral–colour-associated congruent means) facilitation components as a function of response times (Panel A), maximal mouse deviation (Panel B) and partial error rates (Panel C).** The magnitude and 99% CI of each component were estimated by contrasting the marginal means (i.e., marginal effects) of two types of stimuli computed from a fitted linear mixed model.

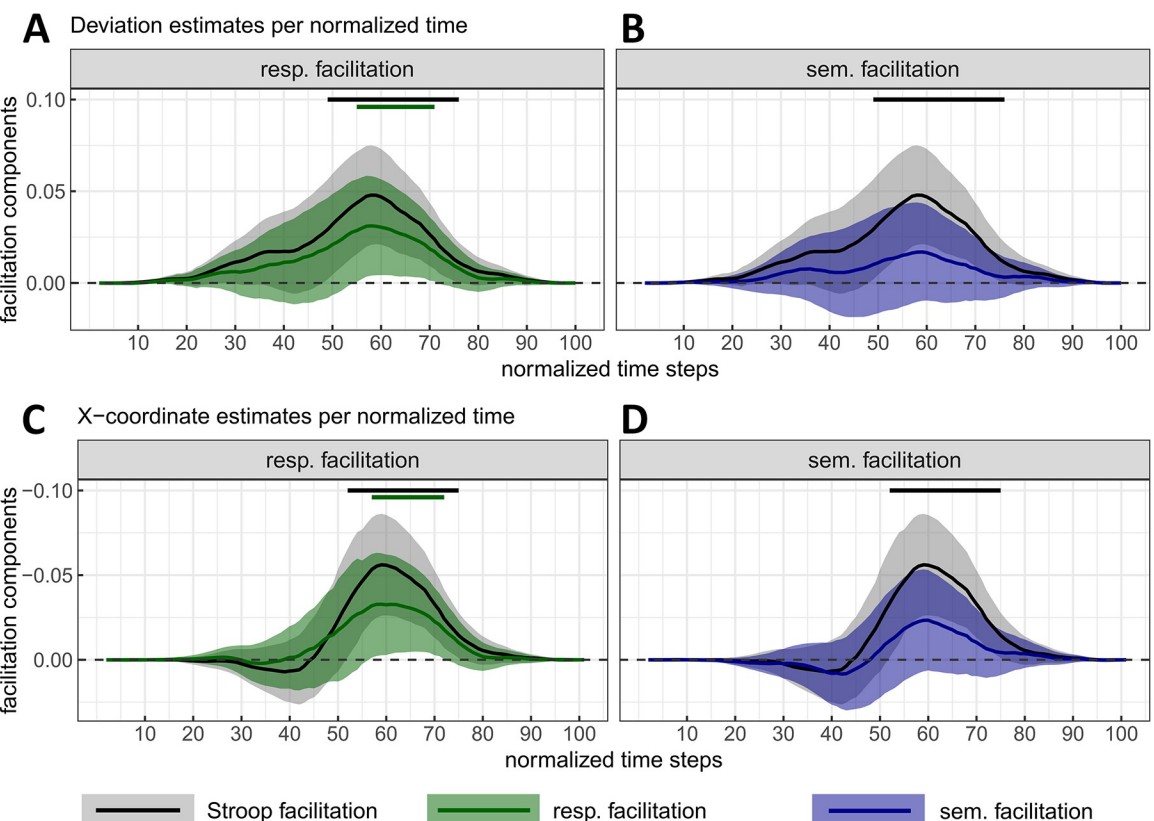

**Fig 11. Stroop facilitation with response (Panels A-C) and semantic facilitation (Panels B-D) components as a function of normalized time.** These are estimated from the mouse deviation towards the incorrect side (A-B) and the X mouse coordinate (C-D). Stroop facilitation is displayed in all plots for visual comparison purposes. Negative X-coordinates indicate a deviation towards the incorrect response (plot's y-axis is inverted for consistency with the other plot). The magnitude and 95% CI (shaded areas) of each component were estimated by contrasting the marginal means of two types of stimuli computed from a fitted linear mixed model (one LMM /time step and /measure, see main text or Fig 10 caption for a detail of the contrasts). Top straight lines indicate the time steps where zero is not included in the 95% CI for at least 5 time-steps (in the main text, only intervals of at least 10 time-steps are reported).

with credible values ranging from -0.6ms to 31.9ms (*99% CI*, *df* = 15366.7; see Fig 10A), and therefore not distinguishable from 0.

Standard colour-congruent items also reduced the deviation by 0.032 (*SE* = 0.015) compared to colour-associated congruent items, with credible values ranging from 0.005 to 0.07 (*99% CI*, *df* = 15288.5; see Fig 10B). This reduction reached its peak at time step 58 with a magnitude of 0.031 (*SE* = 0.014, *95% CI* = [0.004, 0.058], *df* = 15367.3; see Fig 11A). The corresponding confidence interval excluded zero over 17% of movement duration (time steps 55 to 71). In the x-coordinate, response facilitation reached a peak of -0.033 (*SE* = 0.015, *95% CI* = [-0.062, -0.004], *df* = 15367.2) at time step 61. From time steps 57 to 72 (i.e., over 16% of total movement duration), the confidence interval excluded zero (see Fig 11C). In line with the overall facilitation (see above), these changes in mouse deviations were unlikely due to changes in PE rates. Indeed, its average magnitude of 1.73% (*SE* = 1.39, *99% CI* = [-1.87, 5.33], *df* = 410) was indistinguishable from zero (Fig 10C).

Semantic facilitation could be reliably estimated only in RT and response facilitation could not be distinguished from 0 in RT, MD and PE rates. Overall, the mouse-tracking task was not sensitive enough for decomposing reliably Stroop facilitation, although it is important to note that our correction for multiple comparisons on those estimates is quite conservative. Indeed,

response facilitation was reliably non-null over more than 10 consecutive normalized time steps in mouse deviation and x-coordinates, i.e. over 17% and 16% of total response time respectively (using a more powerful confidence interval of 95%). Therefore, those results provide some modest evidence for the multi-stage models of the Stroop effect, but impaired by the low sensitivity to facilitation effects. Additionally, it should be remembered that these two components of Stroop facilitation were not estimated independently (i.e., by using different, and therefore statistically independent, baselines), unlike in the case of the conflicts underlying Stroop interference. Thus, the higher the true value of one component, the lower the true value of the other component is (the sum of the two components equals the Stroop facilitation). In sum, statistically speaking, some uncertainty remains concerning a credible division of Stroop facilitation into these components in the mouse-tracking Stroop task.

## Discussion

By forcing response selection into response execution processes, the present mouse-tracking study investigated: 1) the ongoing process of response selection in the colour-word Stroop task, and; 2) the composite nature of Stroop effects. Response times, maximum mouse deviations, their evolution across normalized time, and partial errors were used to this end.

### Response selection vs. execution processes in the colour-word Stroop task

In real-life contexts, when a response must be initiated quickly, a final decision might not have been taken as to the nature of the ideal response (e.g., heading in one of two possible directions on a bifurcated path, or beginning to answer a question before your mind is made up about what you are going to say). In selective attention tasks such as in the Stroop task it is often not possible to change one's mind once a response has been committed to (e.g., under speeded conditions). In such cases, one might rely on weaker evidence to initiate a response before the final response is selected. Hand-tracking studies of the Stroop effect largely reflect this idea by showing that response selection can continue to evolve after a response has been initiated. Indeed, whether [18] or not [14–17] speeded responding is introduced, the ongoing hand movement continues to deviate from the direct path towards a correct response. However, the extent to which these deviations genuinely reflect the co-activation of competing responses (i.e., prime indicator of the ongoing response selection) remained unclear. Indeed, hand-movements simply directed towards the centre of the screen–that are not infrequent (i.e., ~9.6% if only cluster Cl5 is taken into account, 21.1% if both Cl5 and Cl6 are included, see Fig 3)–also generate movement deviations. By reporting that ~27.1% (see Fig 3) of all considered mouse trajectories were specifically directed towards the incorrect response but ended their course at the correct response (i.e., partial errors [19]), the present study attempted to address this pending issue.

Since hand-tracking studies with movement initiation deadlines encourage partial errors, the added value of this type of analysis might be seen as limited. However, while some caution is undeniably needed, we are inclined to consider partial errors as reflecting the temporal alignment of processes pertaining to response selection vs. response execution. Indeed, for a mouse movement to be corrected, response selection must continue after response initiation towards the incorrect response. This idea is reinforced by results from other response modalities. So-called lexical errors observed with vocal responses (i.e., utterances like "blll. . .green" for $BLUE_{green}$) are indeed consistent with this idea of temporal alignment which is perhaps most directly documented by measures of electromyographic (EMG) activity [56] (see also [57] for a similar analysis in a Stroop-like task). Specifically, measures of sub-threshold EMG activity of the hand that is not supposed to respond suggest that the co-activation of competing

responses can be observed at the level of the effectors (as evidenced by reports of double hand EMG-activations preceding a correct answer) even when standard button presses are used to respond (see also [58–60]). These different results–including partial errors reported in the present study–therefore clearly run against current models of the Stroop effect assuming that response selection and response execution are sequential processes and that all conflict in the Stroop task is resolved at the level of response selection (i.e., strictly before response execution) [4–9, 29].

Hand tracking (mouse- and reach-tracking) studies can further enlighten the ongoing debate about the locus at which Stroop effects (response selection vs. execution) in several possible ways. This could be done by manipulating the movement initiation and response deadlines used in the present and in Bundt et al.'s study orthogonally [18], by examining the extent to which response selection influences movement monitoring [61–65] or by inspecting movement dynamics on response selection through action costs and commitment [66, 67]. Note that the manipulation of the movement initiation deadline will allow the investigation of a more general issue–raised by the absence of differences in initiation times between different conditions (although see [18] for a similar result) and by the rather large partial error rates–that participants are performing "Go before knowing the goal" type [68] of Stroop task that might explain the overlap between response selection and execution. Therefore, future studies should assess whether the Stroop effects are still observable on partial errors when no such deadline is imposed.

## The composite nature of Stroop effect

As anticipated, response times and the mouse deviation measures provided converging evidence that the total Stroop (or congruency) effect–reported in the existing mouse-tracking studies–resulted not only from the interfering effect of standard colour-incongruent items but also the facilitative effect of standard colour-congruent items. Indeed, the use of colour-neutral baseline in the present study revealed that the overall (congruency) Stroop effect of ~159 ms in response times encompassed ~40ms-large facilitation. Despite resulting from much longer response times than those observed with a more standard manual response modality (i.e., with key-pressing), these magnitudes are in line with other past Stroop studies. For instance, in Augustinova et al. [39], an overall Stroop (congruency) effect of ~174 ms, observed with vocal responses, encompassed ~124ms-large interference and ~50ms-large facilitation.

This latter phenomenon was also clearly evidenced in deviations measures (except partial errors). To illustrate, on maximum mouse deviations, facilitation was estimated to explain 21.5% of the total Stroop effect. It should be noted however, that by design, mouse- and reach-tracking are more relevant for estimating interference than facilitation since mouse deviations involve floor effects (i.e., due to the lower bound imposed by the ideal trajectory from start to correct response, deviations cannot be reduced as much as they can be increased). Indeed, the ensuing limitation of the present study is that facilitation was estimated with less sensitivity than interference. Therefore, although the absence of facilitation in partial errors is consistent with the fact that neither colour-congruent nor colour-neutral items are expected to activate a competing (i.e., incorrect) colour-response, it still could simply reflect the latter statistical issue.

In line with remaining challenges in hand-tracking studies of the Stroop effect (see [15] for discussion), the present study also examined interference and facilitation at both the level of stimulus and of response. While response times and mouse deviation measures (except partial errors) provided converging evidence for reliable contributions of response facilitation, semantic facilitation was only credibly seen in response times. If semantic facilitation is indeed

qualitatively different from response facilitation (i.e., it is conceptual in its nature; see [30–32, 34]), it is not surprising that it was not reflected credibly in mouse deviation measurements. But this absence could also be explained rather simply by the aforementioned issue of floor effects and/or by the lack of power for estimating confidently effects that are of small magnitude.

As expected, a reliable contribution of both response and semantic conflict to the overall Stroop interference effect was for the first time also found in different mouse-tracking measures. Although the contribution of semantic conflict was not reliably observable in response times (see Sharma & McKenna [43] for this absence with other types of manual responses), in mouse deviations for instance, semantic conflict explained 23.5% of Stroop interference (see e.g., Augustinova & Ferrand [25] for a similar estimate). In line with the idea that co-activation of two competing response is the most likely to occur for items inducing response conflict, the majority of partial errors occurred for standard colour-incongruent items. However, somewhat contrary to the idea that response and semantic conflicts are qualitatively different, associated colour-incongruent items (items that are only expected to generate semantic conflict, see Introduction section and [25] for discussion) also generated partial errors–with a higher rate than observed for colour-neutral baseline items. Therefore, it is difficult to attribute this small, but reliable semantic conflict in partial errors rates to the aforementioned fact that handing-tracking induces at least some partial errors by design (see previous section). Rather, at least at first glance, these results reinforce single-stage response competition models of Stroop interference. Recall indeed that in Roelofs' model [8], response and semantic-associative conflict are not *qualitatively* different but thought to result from different *quantities* of a single (i.e., response) conflict taking place in the language production unit. Given that the distractor SKY (for $SKY_{yellow}$) is only associated with the response-set colours, it activates the incorrect response (e.g., say "blue"/show blue) less strongly than colour-incongruent items (e.g., $BLUE_{yellow}$), for which distractors (BLUE) are directly included in the response-set, hence explaining present difference in partial errors rates.

Additionally, Roelofs' model could also explain why the mouse-trajectories for incongruent non-response set items ($GREY_{yellow}$) seem to deviate less towards the incorrect response side than those for colour-associated incongruent items $SKY_{yellow}$–an inverse semantic relevance effect that is reinforced by the observation of partial errors rates (but note that this difference is only reliably estimated in x-coordinates, see Fig 9C). Indeed, because the distractor SKY is directly associated with blue (i.e., a colour that is part of the response set), it could therefore interfere more with the production of the correct response (i.e., yellow) than the distractor GREY (that is not part of the response set). If, however this is true, then this model, exactly like other single-stage response competition models, fails to efficiently account for standard semantic relevance effect (i.e., an effect going to the opposite direction) observed in response times in the present study. Indeed, in line with past chronometric studies [37, 43, 45], RTs were longer for non-response set items ($GREY_{yellow}$) than for colour-associated incongruent items ($SKY_{yellow}$). In sum, future studies need to replicate this inverse pattern between RTs and deviation measures further. Still, in our study, this dissociation can simply indicate the difference in the noise level between the two measures. However, if this dissociation between chronometric and deviation measures is further replicated, it is then clearly better explained in terms of two qualitatively distinct conflicts anticipated by multi-stage models of Stroop interference. Indeed, single stage response competition models cannot account for this dissociation.

While colour-associated incongruent items are likely to induce semantic conflict, they also induce at least some *response* conflict [69] (see also [23] for further discussion)–as suggested by partial errors. Non-response set items might therefore seem better candidates for measuring

semantic conflict in future studies. Still, the conflict they generate can be accounted for by the aforementioned single-stage response competition models of the Stroop effect which are therefore most unambiguously contradicted when semantic conflict is induced by so-called same-response trials. Indeed, these items (e.g., $BLUE_{red}$) included in the two-to-one Stroop paradigm [26] generate interference despite the fact their word- and colour-dimension converge towards the same response (i.e., do not involve any response conflict since responses to blue and red are mapped on the same response key, see [23] for an ample discussion of this issue and [70, 71] for empirical evidence). Therefore, future hand-tracking studies of interference at both stimulus and response levels might consider using this paradigm.

This latter type of study also calls for large sample sizes (both in participants and number of trials per conditions) to render analyses of partial errors meaningful. Indeed, superior partial error rates on standard-incongruent trials (also called different response trials) compared to same-response and colour-neutral trials observed in tandem with a linear effect on response times (including longer RTs for same-response than for colour-neutral trials) would unambiguously argue that interference induced by semantic conflict is *qualitatively* distinct than the one induced by response conflict. Because all the components of Stroop interference observed in the present study occurred after a response had been initiated (see also [18] for this type of result), they need to be replicated further under different experimental conditions, namely when no instructions to initiate mouse-movements quickly (used in the present experiment) are provided. This will allow future hand-tracking studies to examine the extent to which semantic as opposed to response conflict influences the threshold adjustment process [14, 15] and/or decision variable build-up [63] along their respective magnitudes in the initiation times.

Despite possible floor effects discussed above, it seems also worthwhile to extend these investigations to semantic (as opposed to response) facilitation. Because neither same response items in the two-to-one Stroop paradigm [26] nor non-response set items have a facilitative counterpart, semantic associates used in the present study still represent a potentially useful measure of performance in the Stroop task despite its aforementioned limitations.

In addition to these and other limitations outlined above, future studies need to extend the examination of different types of facilitation and of interference (stimulus vs. response)–reported in the present study–to the relationship between the current and previous trials. This should be done while also controlling for the extent to which associative priming effects contribute to results reported above. Also, and importantly, the extent to which the pairing of stimuli also contributed to the magnitude of effects observed in the present study is at this point unclear (see [15] for ample discussion of these issues).

## Concluding remarks

Despite these different limitations, the results reported above provide substantial evidence that the overall Stroop interference effect is indeed a composite, rather than a unitary phenomenon, whereas for Stroop facilitation, a composite effect was only seen in response times. Because the dominant single-stage response competition models [4–9] currently fail to explain this composite nature of both facilitation and interference effects, the findings reported in the present study clearly show that mouse- or hand-tracking can significantly add to this ongoing debate [23]. Indeed, mouse- or hand-tracking is especially informative with regard to where Stroop effects are resolved and can be observed. As such it can also contribute to another still-open issue of partial [12, 14–17] or even total overlap [18] between response selection and execution in the Stroop task. Since the only available multistage conflicts model [10] also fails to account for this latter result, the present study calls for further theoretical endeavours in the Stroop

literature to account for the potential for cascading Stroop effects from response selection to response execution. It also calls for a more sophisticated consideration of baselines used to measure Stroop effects [23] and further consideration of where Stroop effects (originating from multiple levels of processing) can be measured and resolved.

## Supporting information

**S1 File. Linear mixed modelling method.** Description of the data analysis method with linear mixed models, including the model selection procedure.
(PDF)

**S2 File. Linear mixed models' fixed and random fitted parameters.** Complete report of the fitted linear mixed models' parameters.
(PDF)

**S3 File. Table of average error rates and time outs (omissions) per conditions.** Report the error rates and non responses rates for each experimental condition.
(PDF)

**S4 File. Clustering method for estimating partial error rates.** Procedure for clustering the mouse trajectories into partial error clusters and non partial error clusters, and for estimating the partial error rates based on these clustering methods.
(PDF)

**S5 File. Statistical analyses of summary variables using ANOVAs.** Replication of the results with ANOVAs and paired comparisons instead of linear mixed models.
(PDF)

**S6 File. Analysis of within and between-participants variability.** An evaluation of the individual differences in the Stroop components as well as a split-half reliability test of the within-participant variability. Unfortunately, the present study not being tailored for it those analyses are underpowered.
(PDF)

## Author Contributions

**Conceptualization:** Boris Quétard, Nicolas Spatola, Ludovic Ferrand, Maria Augustinova.

**Data curation:** Boris Quétard.

**Formal analysis:** Boris Quétard.

**Funding acquisition:** Benjamin A. Parris, Ludovic Ferrand, Maria Augustinova.

**Investigation:** Boris Quétard, Nicolas Spatola.

**Methodology:** Boris Quétard, Nicolas Spatola, Ludovic Ferrand.

**Project administration:** Ludovic Ferrand.

**Resources:** Ludovic Ferrand.

**Software:** Boris Quétard.

**Supervision:** Benjamin A. Parris, Ludovic Ferrand, Maria Augustinova.

**Visualization:** Boris Quétard.

**Writing – original draft:** Boris Quétard, Benjamin A. Parris, Maria Augustinova.

**Writing – review & editing:** Boris Quétard, Nicolas Spatola, Benjamin A. Parris, Ludovic Ferrand, Maria Augustinova.

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
