## [Decision Letter · Decision Letter 0]

8 Aug 2022

PONE-D-22-15842A mouse-tracking study of the composite nature of the Stroop effect at the level of response execution.PLOS ONE

Dear Dr. Quétard,

Thank you for submitting your manuscript to PLOS ONE. I have sent your manuscript to two experts on the topic of mouse tracking and stroop tasks, and I have read the paper myself. I would like to take this opportunity to thank the reviewers for their time and commitment. As you will see from the comments appended below, the reviewers made excellent and constructive comments to improve your manuscript. They believe that the question addressed is timely and that the paper will make a fine contribution to the literature after some revisions. I agree with their assessment and therefore I am inviting a revision. Although the reviewers suggested minor revisions, I am classifying it as major so that you have time to carefully address all the points raised in the review process. Please take in consideration all the points raised and try to address them to the best of your abilities. I will likely send the revised paper again for assessment by the two original reviewers.I also identified a few minor points that I would like to see addressed in the revision:a. There are too many measures detailed in the introduction/method (total stroop, stroop interference, response interference, etc). It is difficult to follow their meaning. I believe some sort of table or even figure that would explain the measures would improve the readability of the paper.

b. The caption of Figure 1 states that there would be a dotted line in the figure. This is not visible in the figure, nor is the d and the explained measures.

c. Missing sample size justification.

d. I believe PE was not defined in the text.

e. Figure panels should be labeled (A, B, C…) instead of simply saying middle, left, right.

f. Lines 606-608: the sentence starts with “Despite…”, but ends without stating the conflicting information. Revise sentence.

g. I was also impressed with the relatively large stroop interference effect observed here. There has been an ongoing debate about the reliability of stroop measures. So I wanted to ask the authors to include a measure of the reliability of their difference scores for RT. If possible, I would also encourage them to compute reliability for the mouse-tracking measures. This would provide important information for future individual difference studies.

We look forward to receiving your revised manuscript.

Kind regards,

Alessandra S. Souza, Ph.D.

Academic Editor

PLOS ONE

Journal Requirements:

“The analysis and interpretation of data; the writing of the report; and the process related to submission of this article for publication was supported by ANR Grant ANR-19-CE28-0013 and RIN Tremplin Grant 19E00851 of Normandie Region, France awarded to senior authors (MA, LF & BP).”

4. Please expand the acronym “ANR, RIN” (as indicated in your financial disclosure) so that it states the name of your funders in full.

Reviewers' comments:

Reviewer's Responses to Questions

**Comments to the Author**

1. Is the manuscript technically sound, and do the data support the conclusions?

Reviewer #1: Partly

Reviewer #2: Yes

2. Has the statistical analysis been performed appropriately and rigorously? 

Reviewer #1: N/A

Reviewer #2: Yes

3. Have the authors made all data underlying the findings in their manuscript fully available?

Reviewer #1: Yes

Reviewer #2: Yes

4. Is the manuscript presented in an intelligible fashion and written in standard English?

Reviewer #1: Yes

Reviewer #2: Yes

5. Review Comments to the Author

Reviewer #1: The present study addresses the dissociation of semantic/response conflicts of Stroop effects. Replacing the conventional reaction-time paradigm, the authors use the mouse-tracking technique to assess the conflict/facilitation effects at both decision and execution levels. The experimental design is well-thought-out; the results, particularly the analyses of partial errors, are inspiring. Meanwhile, some limitations of the current research have been discussed.

Before this paper is suitable for publication, several issues need to be addressed.

1. One of the main claims in this paper is that mouse-tracking helps reveal the potential overlap between semantic decision and response execution of Stroop effects, as shown by features of mouse trajectories such as partial errors. However, given the nature of the reaching task per se, there are always more or less partial errors observed, since reversing the direction halfway during reaching is often seen in mouse-tracking studies. Thus, it seems farfetched to interpret the partial errors as evidence to overlap of cognitive processes in a specific task (i.e., Stroop) in the current study. Even if previous research with different paradigms suggests that the overlap could exist, it is unclear whether the partial errors in the current study stem from the overlap of selection/execution or other underlying cognitive processes. The authors may want to discuss this limitation further and be more careful about the conclusion.

2. For those Stroop effects examined, the authors report the 95% CI, which corresponds to a p-value criterion of 0.05. However, since there are multiple comparisons conducted, the risk of false-positive is above 5% when 95% CI is repeatedly applied. Thus, p-value corrections for multi-comparisons are needed. Please provide details about p-value corrections, if any. In addition, reporting only 95% CIs is insufficient for readers to evaluate those effects; effect sizes should also be included.

3. In mouse-tracking analysis, there are alternative features of mouse trajectories to assess the deviation. For example, AUC (area-under-the-curve) is another often-used measure for deviation, especially in semantic conflicts (e.g., Xiao & Yamauchi, 2015). Given that the MD and AUC are usually positively correlated, why choose MD over other features (e.g., AUC) as the index of semantic conflicts? The authors may consider reports other relevant measures as well or explain the reason for relying on MD.

4. In lines 204-205, “participants were also asked to identify the font colour of associated (e.g., SKYgreen) non-response set incongruent trials…” How exactly were participants instructed to do with “non-response?” Please clarify. It may cause confusion because some readers may interpret “non-response trials” as being “completely ignored/no response at all.”

5. Make sure the first line of each paragraph is indented in a consistent manner.

Reviewer #2: Classically, delay in response observed in Stroop tasks has been attributed to the conflict in the input processing stage. In this paper, using a mouse tracking task instead of a simple button press, the authors aimed to decompose the different types of conflict processing in the Stroop task which evolves in the response execution phase.

Response times, maximum deviation towards the interfering targets and the partial error estimate were extracted from the cursor trajectory during the response execution. Authors evaluated the Stroop interference effect and facilitation effect using these measures, decomposing them into response conflict/facilitation and the semantic conflict/facilitation components by setting appropriate control conditions. Authors successfully showed the different magnitude of effect induced by these different factors.

The experiment is performed thoroughly, and the manuscript is written clearly. I believe this shows an elegant way to differentiate the factors contributing to the classical Stroop effect, particularly in the response execution phase.

Below are the points which need more clarification.

1) One of the main messages of this paper is that the stroop interference effect is not necessarily solved before the action execution but can evolve during the response execution phase. I do understand that the participants are purposefully instructed to initiate the action quickly to facilitate the decision to overlap with the execution. However, no difference in initiation times between different conditions seems still very strange to me (167ms-170ms). It seems like participants are performing “Go before knowing the goal” type task (Chapman et al., 2010). Please discuss the reason for why there is no difference in the initiation times, and what can be affected by this.

2) If I am understanding correctly, if the velocity is similar, response time and the maximum deviation should correlate. How exactly do they correlate, or do they not? In some cases (semantic conflict), authors found dissociation between the response time and the maximum deviation measures, however, I was wondering what this exactly means; is it simply indicating the difference in the noise level between the two measures or is it pointing to the different processing of velocity and the trajectory interference magnitude. Please discuss about this point.

3) The part describing the analysis was very difficult to understand. For example, for line 327, when the authors say, “above and below direct path”, what is up and what is down? I think it can be explained more clearly, perhaps including one schematic figure for it.

6. PLOS authors have the option to publish the peer review history of their article (what does this mean?). If published, this will include your full peer review and any attached files.

Reviewer #1: No

Reviewer #2: No

---

## [Author Response · Author response to Decision Letter 0]

20 Oct 2022

Dear,

Please find our responses to the Editor's comments and suggestions in the cover letter document and our responses to the reviewers' comments and suggestions in the response to the reviewers document. We uploaded them together with the manuscript files, figures and supporting information.

Best regards.

---

## [Decision Letter · Decision Letter 1]

29 Nov 2022

A mouse-tracking study of the composite nature of the Stroop effect at the level of response execution.

PONE-D-22-15842R1

Dear Dr. Quétard,

We’re pleased to inform you that your manuscript has been judged scientifically suitable for publication and will be formally accepted for publication once it meets all outstanding technical requirements. Thank you for addressing the remaining points in your revision. The reviewers were fully satisfied with the implemented changes.

Kind regards,

Alessandra S. Souza, Ph.D.

Academic Editor

PLOS ONE

Additional Editor Comments (optional):

Reviewers' comments:

Reviewer's Responses to Questions

**Comments to the Author**

1. If the authors have adequately addressed your comments raised in a previous round of review and you feel that this manuscript is now acceptable for publication, you may indicate that here to bypass the “Comments to the Author” section, enter your conflict of interest statement in the “Confidential to Editor” section, and submit your "Accept" recommendation.

Reviewer #1: All comments have been addressed

Reviewer #2: All comments have been addressed

2. Is the manuscript technically sound, and do the data support the conclusions?

Reviewer #1: Yes

Reviewer #2: Yes

3. Has the statistical analysis been performed appropriately and rigorously? 

Reviewer #1: Yes

Reviewer #2: Yes

4. Have the authors made all data underlying the findings in their manuscript fully available?

Reviewer #1: Yes

Reviewer #2: Yes

5. Is the manuscript presented in an intelligible fashion and written in standard English?

Reviewer #1: Yes

Reviewer #2: Yes

6. Review Comments to the Author

Reviewer #1: The authors have addressed all my comments in details. The responses are solid. I am satisfied with the revision.

Reviewer #2: Authors have thoroughly answered to all the reviewer’s comment. I have no further comment for the manuscript.

7. PLOS authors have the option to publish the peer review history of their article (what does this mean?). If published, this will include your full peer review and any attached files.

Reviewer #1: No

Reviewer #2: No

---

## [Editor Report · Acceptance letter]

6 Jan 2023

PONE-D-22-15842R1 

A mouse-tracking study of the composite nature of the Stroop effect at the level of response execution. 

Dear Dr. Quétard:

I'm pleased to inform you that your manuscript has been deemed suitable for publication in PLOS ONE. Congratulations! Your manuscript is now with our production department. 

Kind regards, 

on behalf of

Dr. Alessandra S. Souza 

Academic Editor

PLOS ONE